# Symptoms of Infarction in Women: Is There a Real Difference Compared to Men? A Systematic Review of the Literature with Meta-Analysis

**DOI:** 10.3390/jcm11051319

**Published:** 2022-02-27

**Authors:** Martin Cardeillac, François Lefebvre, Florent Baicry, Pierrick Le Borgne, Cédric Gil-Jardiné, Lauriane Cipolat, Nicolas Peschanski, Laure Abensur Vuillaume

**Affiliations:** 1Emergency Department, Regional Hospital of Metz-Thionville, 57000 Metz, France; martin.cardeillac@hotmail.fr (M.C.); l.cipolat@chr-metz-thionville.fr (L.C.); 2Medical School, University of Lorraine, 9 Avenue de la forêt de Haye, Vandœuvre-lès-Nancy, 54505 Nancy, France; 3Groupe de Méthodes en Recherche Clinique (GMRC), University Hospital of Strasbourg, 67000 Strasbourg, France; francois.lefebvre@chru-strasbourg.fr; 4Emergency Department, Hôpitaux Universitaires de Strasbourg, 67000 Strasbourg, France; florent.baicry@chru-strasbourg.fr (F.B.); pierrick.leborgne@chru-strasbourg.fr (P.L.B.); 5Institut National de la Santé et de la Recherche Médicale—Unité Mixte de Recherché (INSERM UMR) 1260, Regenerative NanoMedicine (RNM), Fédération de Médecine 63 Translationnelle (FMTS), University of Strasbourg, 67000 Strasbourg, France; 6Pole of Emergency Medicine, University Hospital of Bordeaux, 33000 Bordeaux, France; cedric.giljardine@chu-bordeaux.fr; 7Equipe AHEAD (Assessing Health in a Digitalizing Real World-Setting), Bordeaux Population Health Research Center Inserm U1219, Universite de Bordeaux—ISPED, 146 rue Léo Saignat, 33076 Bordeaux, France; 8Unité Mixte de Recherche Epidémiologique et de Surveillance Transport Travail Environnement (TS2-UMRESTTE), Université Gustave Eiffel, Campus de Lyon, 25 Avenue François Mitterrand, Case 24, 69675 Bron, France; 9Emergency Department, University Hospital of Rennes, 35000 Rennes, France; nicolas.peshanski@chu-rennes.fr; 10Rennes-1 University School of Medicine, 35000 Rennes, France; 11International Research Lab 2958 Georgia Tech Institute—Centre National de la Recherche Scientifique (CNRS), 57000 Metz, France

**Keywords:** acute coronary syndrome, cardiovascular disease, gender-based difference, women

## Abstract

(1) Context: The management of acute coronary syndrome (ACS) is based on a rapid diagnosis. The aim of this study was to focus on the ACS symptoms differences according to gender, in order to contribute to the improvement of knowledge regarding the clinical presentation in women. (2) Methods: We searched for relevant literature in two electronic databases, and analyzed the symptom presentation for patients with suspected ACS. Fifteen prospective studies were included, with a total sample size of 10,730. (3) Results: During a suspected ACS, women present more dyspnea, arm pain, nausea and vomiting, fatigue, palpitations and pain at the shoulder than men, with RR (95%CI) of 1.13 [1.10; 1.17], 1.30 [1.05; 1.59], 1,40 [1.26; 1.56], 1.08 [1.01; 1.16], 1.67 [1.49; 1.86], 1.78 [1.02; 3.13], respectively. They are older by (95%CI) 4.15 [2.28; 6.03] years compared to men. The results are consistent in the analysis of the ACS confirmed subgroup. (4) Conclusions: We have shown that there is a gender-based symptomatic difference and a female presentation for ACS. The “typical” or “atypical” semiology of ACS symptoms should no longer be used.

## 1. Introduction

Acute coronary syndrome (ACS) is one of the leading causes of death worldwide. It is responsible for 7.4 million deaths annually, particularly in developed countries [1,2]. The management of ACS is based on a rapid diagnosis in order to lead the patient to an adequate place of support and to provide him the right treatment [3,4]. The faster the diagnosis, the lower the risk of death or complications. If the final diagnosis of ACS is based on clinical signs of myocardial ischemia, such as chest pain, electrocardiographic signs (ST-Elevation Myocardial Infarction, STEMI; or Non-ST-Elevation Myocardial Infarction, NSTEMI), and on cardiac biomarkers, such as increased troponine [3], its semiology is very heterogeneous. Thus, while it is clear that chest pain is the main symptom, other symptoms such as dyspnea, digestive disorders and fatigue are described as “atypical” [5]. This “atypical” designation has been called into question for several years, and today it is rather thought that the symptomatology depends on the patient’s gender [5].

Lately, several authors were interested in the ACS differences between men and women [6,7,8,9,10,11,12,13,14,15,16,17,18,19,20,21,22,23]. First of all, physiopathologically, men and women are not exposed to cardiovascular risk factors in the same way. In fact, estrogen provides some cardioprotection, but exposure to tobacco is more harmful in women than in men [24]. The clinical presentation is also reported to be different, men being described as presenting classic symptoms such as chest pain, whereas women can present more heterogeneous symptoms close to the form historically described as “atypical” [25,26]. Women tend not to receive treatment or to receive sub-optimal treatment for ACS due to the late recognition of the acute event [27].

In this context, the main goal of our systematic review of the literature with a meta-analysis was to focus on the ACS symptoms differences according to gender in order to contribute to the improvement in knowledge regarding the clinical presentation in women.

## 2. Materials and Methods

### 2.1. Study Design

We carried out a systematic review of the literature and then a meta-analysis of these studies using the “Preferred Reporting Items for Systematic Reviews and Meta-analyzes” (PRISMA) methodology [28]. We searched two databases, PubMed and COCHRANE, with the aim of studying the articles published between 1 January 2010 and 4 February 2021. We included analyses focused on the clinical symptoms of ACS according to gender (the search equations are available in Figure 1). All prospective, retrospective, observational and interventional studies were included. The exclusion criteria were the unavailability of the entire text (only abstract available despite an extensive search on other existing databases), the absence of information on the suspected or retained diagnosis of ACS, the absence of any distinction between men and women, the absence of a description of symptoms and case reports. We excluded all studies that did not meet at least one quality criterion described below. Studies published in a language other than French or English were also excluded.

Two co-authors (M.C. and L.A.V.) independently assessed the studies for eligibility. Differences were resolved by consensus.

### 2.2. Data Collection and Processing

Two coauthors (M.C. and L.A.V.) independently extracted data from the included full-text citations with a report form validated by N.P. The following information was abstracted: the last name of the first author, publication year, country where the study was performed, study design, total participants in the study, numbers of male and female participants, mean age of each sex, number of patients and percentage of symptoms of ACS (chest pain, dyspnea, arm pain, sweating, nausea and vomiting, fatigue, palpitations, shoulder pain, abdominal and epigastric pain). Data were collected with Excel^®^ (Microsoft Corporation, Richmond, VA, USA).

### 2.3. Quality Assessment

We used an adapted version of the Newcastle Ottawa Scale [29]. We awarded 4 stars for the “selection” yardstick to studies that used random or consecutive selection of their patients. For the “comparability” yardstick, we awarded 2 stars to studies that adjusted their results with multiple covariates. Finally, the independent analysis of symptoms allowed the attribution of one star for the “outcomes” yardstick.

### 2.4. Statistical Analysis

The outcomes measured in studies of sex differences in the symptom presentation of patients with suspected or confirmed acute coronary syndrome were binary covariates. For these binary outcomes, the risk ratio was used to measure sex’s effect on the symptom presentation. The overall estimate was taken from a fixed effects or a random effects model for heterogeneity testing using a standard chi-square statistic. A subgroup analysis of patients with a diagnosis of ACS confirmed by the authors was performed. All analyses were conducted using R software version 4.0.2. A value of *p* < 0.05 was considered significant for main effect and heterogeneity tests. 

Heterogeneity was first taken into account by running a random effects model when necessary. Then, a metaregression with the region was carried out to analyze its effect on the logarithm of the relative risk.

## 3. Results

### 3.1. Characteristics of Included Studies

We obtained 5403 results on the “men” equations, 4981 results on the “women” equations and 346 results on the “comparison” equations, for a total of 10,730 articles. After an initial screening of the titles of the articles and then the abstracts, followed by a review of the full texts and the application of the inclusion and exclusion criteria, 96 studies were selected. After removing duplicate articles, and excluding studies that did not contain data that could be used in a meta-analysis (non-individual statistics, meta-analysis), 15 studies were approved (Figure 2) for the meta-analysis studies, including all prospective studies.

Baseline characteristics of the 15 included studies are summarized in Table 1. Heterogeneity was first taken into account by running a random effects model.

### 3.2. Patients’ Characteristics

Our study included 1,213,709 patients: 711,149 men (58.59%) and 502,560 women (41.41%). The “confirmed ACS” subgroup represented 1,195,524 patients, including 494,641 women. Patients included in the studies were admitted to emergency departments for chest pain, ACS suggestive symptoms according to the receiving physicians [10,12,16,17,18,20,21] or ACS suspicion from troponin measurement [11,13], or were hospitalized for ACS [9,14,15,19,22,23].

### 3.3. Symptoms

During an ACS, women are more likely to have dyspnea than men (RR (95%CI): 1.13 [1.10; 1.17]), arm pain in either arm (RR (95%CI): 1.30 [1.05; 1.59]), nausea and vomiting (RR (95%CI): 1.40 [1.26; 1.5]) and fatigue (RR (95%CI): 1.08 [1.01; 1.16]) (Table 2 and Figure 3). They are also more likely to present with palpitations (RR (95%CI): 1.67 [1.49; 1.86]) and pain in the shoulder (RR (95%CI): 1.78 [1.02; 3.13]) (Table 2 and Figure 3). Finally, the women were older by 4.15 years compared to men (95%CI): 4.15 [2.28; 6.03]) (Table 2 and Figure 3).

The results are consistent in the analysis of the ACS confirmed subgroup. All results are available in Table 2, Figure 3 and Figure 4 and in Appendix A.

There was an effect of the region on the logarithm of the relative risk only for nausea-vomiting, increased or irregular heart rate and shoulder pain with all patients.

## 4. Discussion

Our systematic review of the literature, with meta-analysis, on a large volume of patients, confirmed that the clinical presentation of ACS in women is different from that of men, and may include pain in the upper limbs, palpitations, dyspnea, nausea and vomiting or simply fatigue. Women are also older during their first attack on average. To our knowledge, this is the first meta-analysis on the subject. We included 15 prospective studies, using RR calculations and a subgroup study with confirmed ACS. Indeed, previous literature reviews lacked precision due to the missing standardization in data collection [30,31]. The standardization proposed by the ESC in the definition of ACS makes it possible to limit bias in the inclusion of studies [3]. Given the large number of studies on ACS, it was also important to produce a strong synthesis using the robust method of meta-analysis. Our meta-analysis was based on multiple databases, minimizing the possibility that evidence-based studies were missing from the analysis. The heterogeneity between the studies was taken into account with a random effects model when necessary. The studies analyzed took place in several countries, with different health systems, making it possible to include populations from the United States, Europe and Asia (including the Arabian Peninsula). Nevertheless, our main strengths have been to include a significant number of patients (over one million), thanks to the analysis of high quality studies. Indeed, all the studies were prospective and therefore did not rely on a posteriori collection of the symptoms of ACS presented by the patients. Data collection was standardized in each of the studies by systematic collection by the investigators [6,7,8,9,10,11,12,13,14,15,16,17,18,19,20,21,22,23]. Carrying out the analysis of the “Confirmed ACS” subgroup allowed us to analyze the symptoms even better. 

Our results are consistent with those of the meta-analysis by Van Oosterhoutand et al., who also studied presentation differences between males and females [32]. This study, although exhaustive, was limited to people with confirmed ACS and did not compare the gender differences in symptom presentation in people with suspected ACS. The classic symptomatology of ACS, which is chest pain, as known by the general public and taught in medical universities, has a strong gender bias [5,33,34]. ACS in women takes a less classic shape, and these symptoms are poorly known by the medical profession [35]. This notion has been emerging for several years. Thereby, De Von et al. proposed to stop using the terminology “typical” and “atypical” or to specify to which reference group it applies [6]. In this way, our study also supports this position, and all recent data should be widely disseminated so that these terms “typical” and “atypical” disappear completely from semiology. Prevention campaigns intended for the general public are also beginning to describe the different clinical symptoms between men and women [36]. These differences in clinical presentation could be linked to different pathophysiology. It is now known that some forms of ACS are more frequent in women, such as ACS type 2, which is based on an inadequacy in myocardial oxygen caused by insufficient intake, without acute coronary injury; and myocardial infarction with non-obstructive coronary arteries (MINOCA), whose pathophysiology has not been clearly elucidated [3]. 

Unfortunately, all of the studies on ACS in our meta-analysis did not specify the pathophysiology of the latter, which may therefore be predominant in women. Cohorts underlining the mechanisms of ACS could help to clarify the link between the clinical presentation and the gender of the patient and the kind of ACS.

There was an effect of the region on the logarithm of the relative risk only for nausea-vomiting, increased or irregular heart rate and shoulder pain with all patients. There is sometimes significant heterogeneity in some symptoms, which is why the models used were random effects. The meta-regression showed that the heterogeneity could be partly explained by the region, but other factors are also possibly responsible.

### Limitations

Our study had several limitations. First, not all territories were covered, especially not low-income countries, thereby limiting the generalization of our results to middle- or high-income countries. Secondly, our equations excluded unpublished studies in English and French. Some studies, published in other languages, may potentially find different results. Thirdly, some symptoms were lacking in precision—for example, the lateralization of pain for pain in the arm or shoulder. All the selected studies reported the symptom of chest pain; however, it was not possible to analyze this symptom for two studies because chest pain was divided into several locations (retrosternal, laterothoracic, etc.) and we could not determine how many patients had chest pain. Then some symptoms, such as confusion and headache, which were mentioned in some studies, could not be included in our meta-analysis due to a lack of data.

## 5. Conclusions

Women with ACS are at greater risk of developing clinical symptoms such as dyspnea, arm pain, nausea, vomiting, fatigue and palpitations than men. In light of this knowledge, the notion of “typical” or “atypical” ACS symptoms should no longer exist. More than half of the victims of ACS are women, so it is now more than necessary to give this knowledge to health students, health professionals and the general public.

## Figures and Tables

**Figure 1 jcm-11-01319-f001:**
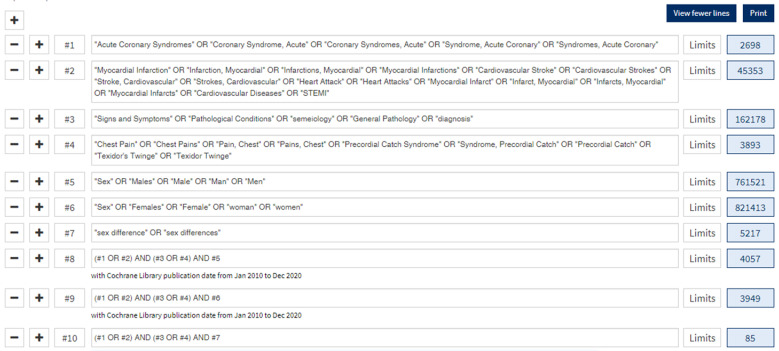
Research equation from Cochrane.

**Figure 2 jcm-11-01319-f002:**
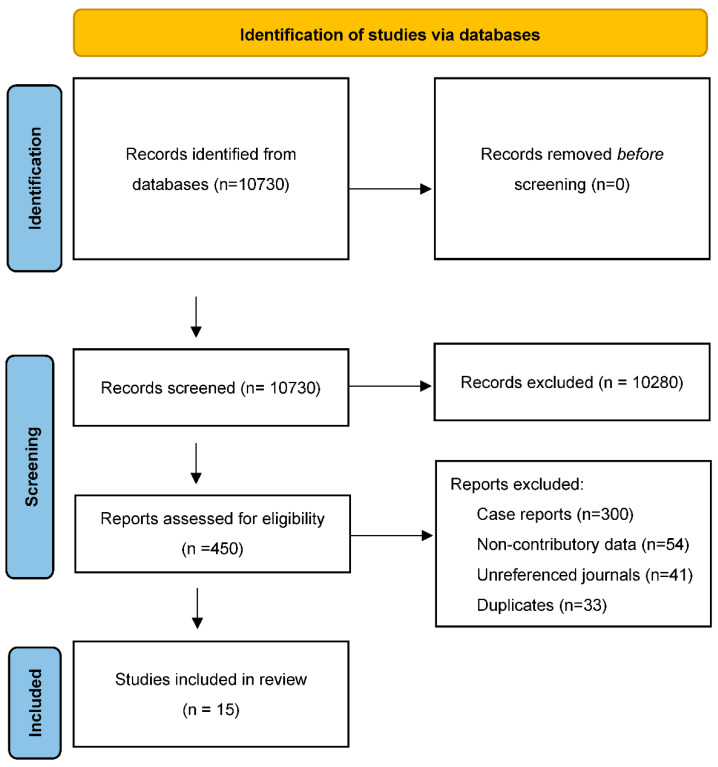
Flowchart.

**Figure 3 jcm-11-01319-f003:**
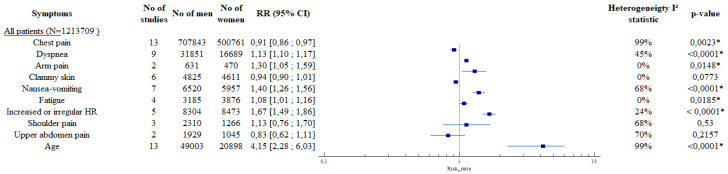
Results of the meta-analysis, forest plots. Legend: * = *p* < 0.005.

**Figure 4 jcm-11-01319-f004:**
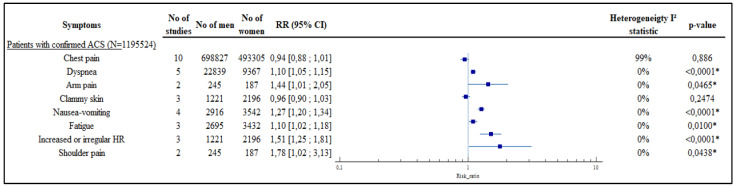
Results of the meta-analysis, subgroup of confirmed ACS, forest plots. Legend: * = *p* < 0.005.

**Table 1 jcm-11-01319-t001:** Summaries of included studies.

Author, Year Published, Journal	Impact Factor 2018	Adapted Newcastle-Ottawa Scale	Region, State, Country	Study Group	Design	ACS Definition for Inclusion	Sample Size	Men	Women
Bjerking, 2016, BMC Cardiovasc Disord [9]	1.947	4 + 1 = 5	Denmark	4000 patients admitted with first AMI 250 females randomly selected; 250 males matched on age and availability of cardiac invasive facilities in the index hospital. 1 patent excluded because of no access to medical record	Matched cohort study	ICD 10th edition codes for ACS I21-I21.9	499	249	250
Canto, 2012, JAMA [10]	51.273	4 + 2 = 6	U.S.A.	1977 hospitals who participated at the National Registry of Myocardial Infarction (NRMI)	Observational Study	ICD 9th edition 410.X1 OR supporting evidence of MI (elevated cardiac biomarker level, electrocardiographic evidence of ACS, or alternative enzymatic, nuclear cardiac imaging, or autopsy evidence indicative of ACS)	1,143,513	661,932	481,581
DeVon, 2014, J Am Heart Assoc [11]	4.66	4 + 2 + 1 = 7	U.S.A.	Patients admitted in 4 large medical centers: 1 in the Midwest, 2 in the Pacific Northwest, 1 in the West region of the United States	Prospective Observational Study	ECG criteria (new ST elevation at the J-point > 0.1 mV in 2 contiguous leads and/or new horizontal or down-sloping ST depression > 0.05 mV in 2 contiguous leads and/or T inversion > 0.1 mV in 2 contiguous leads with prominent R wave) and/or troponin criteria (outside the referenced norm for the institution)	736	464	272
Ferry, 2019, J Am Heart Assoc [12]	4.66	4 + 2 = 6	Scotland	Patients admitted in the ED of the Royal Infirmary of Edinburgh for suspected ACS	Substudy of a prospective trial	requested cardiac troponin for suspected ACS	1941	1185	756
Gimenez, 2014, JAMA Int Med [13]	20.768	4 + 1 = 5	Switzerland, Spain, Italy	Patients admitted in 9 study centers who participated at the Advantageous Predictors of Acute Coronary Syndrome Evaluation (APACE) study, with symptoms suggestive of AMI	Prospective Observational Study	acute chest pain with an onset or peak within the last 12 h	2475	1679	796
Lee, 2019, J Am Col Cardiol [14]	18.639	4 + 2 = 6	Scotland	All patients with suspected ACS who presented in the 10 participating hospitals of the High-Sensitivity Troponin in the Evaluation of Patients	Stepped-wedge, cluster-randomized controlled trial	suspected acute coronary syndrome and had paired troponin measurement with the contemporary and the trial assay	10,360	5369	4991
Lichtman, 2018, Circulation [15]	23.054	4 + 2 + 1 = 7	U.S.A.	Patients hospitalized with AMI in 103 hospitals participating in the Variation in Recovery: Role of Gender on Outcomes of Young AMI Patients (VIRGO) study	Prospective Observational Study	increased cardiac biomarkers levels AND symptoms of ischemia OR ECG changes indicative of new ischemia (new ST-T changes or development of pathological Q waves)	2985	976	2009
Nanna, 2019, Circ Cardiovasc Qual Outcomes [16]	4.378	4 + 2 + 1 = 7	U.S.A.	Patients hospitalized for AMI, enrolled in the ComprehenSIVe Evaluation of Risk Factors in Older Patients with Acute Myocardial Infarction (SILVER-AMI) study.	Prospective Observational Study	criteria for the Third Universal Definition of acute myocardial infraction	3041	1695	1346
Patel, 2015, Glob Heart [17]	3.238	4 + 2 + 1 = 7	India	ACS admissions in 125 hospitals who participated in the Kerala ACS Registry	Prospective Observational Study	admission with chest pain AND at least one of the following criteria (ST-segment elevation in 2 contiguous leads with ou without reciprocal ST-segment depression OR troponin or creatinin kinase-myocardial band elevation OR ST-segment depression OR T-wave inversion in 2 contiguous leads with an history of coronary heart disease)	25,748	19,923	5825
Pelter, 2012, Am J Emerg Med [18]	1.651	4 + 1 = 5	U.S.A., New Zealand, Australia	Secondary analysis of the Patient Response to Myocardial Infarction following a Teaching Intervention Offered by Nurses (PROMOTION) trial	Secondary analysis of a randomized controlled trial		565	367	198
Shah, 2015, BMJ [19]	27.604	4 + 2 = 6	Scotland	Patients presenting to the Royal Infirmary of Edinburg with suspected ACS	Prospective Cohort Study	suspected an acute coronary syndrome	1126	622	504
Shebab, 2020, J Am Heart Ass [20]	4.66	4 + 2 = 6	Kuwait, Qatar, Bahrain, United Arab Emirates, Oman, Yemen, Saudi Arabia	Patients with a diagnosis of ACS enrolled in one of the 7 Arabian Gulf Registry	Prospective Consecutive Study	standard definition according to published American College of Cardiology/european Society of Cardiology	15,532	13,499	2033
Sörensen, 2018, J Am Heart Ass [21]	4.66	4 + 1 = 5	Germany	Patients presenting to the ED of the University Heart Center Hamburg, enrolled in the Biomarkers in Acute Cardiac Care (BACC) study and in the StenoCardia Study	Prospective Cohort Study	ACS suspected OR acute chest pain	2520	1640	880
Van de Meer, 2015, PLOS ONE [22]	2.776	4 + 2 + 1 = 7	Netherlands	All patient admitted to the cardiac ED with chest pain, enrolled in “the prospective validation of the HEART score”	Prospective Observational Study	chest pain	2331	1328	1003
You, 2018, Aging and Disease [23]	4.232	4 = 4	China	Patients with STEMI admitted in 2 hospitals and undergo to PPCI	Prospective Observational Study	STEMI who underwent PPCI	337	220	117

Legend: PPCI = primary percutaneous coronary intervention; ACS = acute coronary syndrome, STEMI = ST elevation myocardial infraction.

**Table 2 jcm-11-01319-t002:** Results of the meta-analysis. RR of getting the symptom if a woman.

Symptoms	No of Studies	No of Men	No of Women	RR (95%CI)	Heterogeneigty I^2^ Statistic	*p*-Value
All patients (N = 1,213,709)						
Chest pain	13	707,842 0.8528 [0.7998; 0.8936]	500,761 0.7831 [0.7177; 0.8368]	0.91 [0.86; 0.97]	99%	0.0023 *
Dyspnea	9	31,851 0.3113 [0.2060; 0.4405]	16,689 0.3595 [0.2442; 0.4937]	1.13 [1.10; 1.17]	45%	<0.0001 *
Arm pain	2	631 0.2072 [0.1309; 0.3120]	470 0.2736 [0.1743; 0.4020]	1.30 [1.05; 1.59]	0%	0.0148 *
Clammy skin	6	4825 0.2520 [0.1579; 0.3771]	4611 0.2312 [0.1349; 0.3671]	0.94 [0.90; 1.01]	0%	0.0773
Nausea-vomiting	7	6520 0.2283 [0.1502; 0.3312]	5957 0.3277 [0.2266; 0.4478]	1.40 [1.26; 1.56]	68%	<0.0001 *
Fatigue	4	3185 0.2300 [0.0788; 0.5107]	3876 0.2368 [0.0752; 0.5420]	1.08 [1.01; 1.16]	0%	0.0185 *
Palpitations	5	8304 0.0729 [0.0332; 0.1524]	8473 0.1222 [0.0576; 0.2407]	1.67 [1.49; 1.86]	24%	<0.0001 *
Shoulder pain	3	2310 0.0947 [0.0333; 0.2410]	1266 0.1161 [0.0371; 0.3093]	1.13 [0.76; 1.70]	68%	0.5300
Upper abdomen pain	2	1929 0.0575 [0.0282; 0.1137]	1045 0.0632 [0.0499; 0.0796]	0.83 [0.62; 1.11]	70%	0.2157
Age	13	49,003	20,898	4.15 [2.28; 6.03]	99%	<0.0001 *
Patients with confirmed ACS (N = 1,195,524)						
Chest pain	10	698,827 0.8404 [0.7868; 0.8826]	493,305 0.7932 [0.7272; 0.8466]	0.94 [0.88; 1.01]	99%	0.886
Dyspnea	5	22,839 0.3104 [0.1809; 0.4785]	9367 0.3726 [0.2164; 0.5609]	1.10 [1.05; 1.15]	0%	<0.0001 *
Arm Pain	2	245 0.1959 [0.1509; 0.2504]	187 0.3513 [0.2189; 0.5113]	1.44 [1.01; 2.05]	0%	0.0465 *
Clammy skin	3	1221 0.2966 [0.1185; 0.5695]	2196 0.2915 [0.1202; 0.5532]	0.96 [0.90; 1.03]	0%	0.2474
Nausea-vomiting	4	2916 0.2856 [0.1517; 0.4719]	3542 0.3860 [0.2265; 0.5744]	1.27 [1.20; 1.34]	0%	<0.0001 *
Fatigue	3	2695 0.3331 [0.2539; 0.4229]	3432 0.3725 [0.2902; 0.4629]	1.10 [1.02; 1.18]	0%	0.0100 *
Palpitations	3	1221 0.0807 [0.0328; 0.1854]	2196 0.1443 [0.0617; 0.3017]	1.51 [1.25; 1.81]	0%	<0.0001 *
Shoulder pain	2	245 0.1005 [0.0279; 0.3028]	187 0.2107 [0.0541; 0.5545]	1.78 [1.02; 3.13]	0%	0.0438 *

* *p* < 0.005.

## Data Availability

The datasets used and/or analyzed during the current study are available from the corresponding author on reasonable request.

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
