# Peer review of "Symptoms of Infarction in Women: Is There a Real Difference Compared to Men? A Systematic Review of the Literature with Meta-Analysis"

_jcm, 2022, doi:10.3390/jcm11051319_

Round 1

Reviewer 1 Report

This is a well written and well done systematic review and metanalysis on symptoms in women with acute coronary syndromes. Here my comments

Minor and major comments:

- The search terms used have to be reported in the main paper.

  • For outline of the search strategy (figure 1) please use the official Prisma flowchart
  • Use palpitations instead of "inreased or irregular HR"
  • Only patients with confirm ACS diagnosis have to be included in the analysis. Diagnostic criteria used for ACS diagnosis in different studies have to be reported too.

Author Response

#Reviewer 1

This is a well written and well done systematic review and metanalysis on symptoms in women with acute coronary syndromes. Here my comments

Response: We thank the reviewer and hope that the revised version will meet his expectations.

Minor and major comments:

- The search terms used have to be reported in the main paper.

Response: We have integrated figure S1 (now figure 1) in the main paper.

  • For outline of the search strategy (figure 1) please use the official Prisma flowchart

Response: We have modified the figure according to PRISMA recommendation

  • Use palpitations instead of "inreased or irregular HR"

Response: We have corrected this term in the manuscript and in the supplementary material.

  • Only patients with confirm ACS diagnosis have to be included in the analysis.

Response: We performed two analyses: one with all patients and one with patients with a confirmed diagnosis of SCA.

Most articles do not make the distinction, so we did the two separate analyses and felt it was important to do so. However, if the editor wishes, we can put these analyses (unconfirmed SCA) as supplementary material.

Diagnostic criteria used for ACS diagnosis in different studies have to be reported too.

Response: We have added a column to the table.

Reviewer 2 Report

Well done metanalysis and this is crucial. In relation to be an original contribution, nothingis  really new about ACS symptoms in women. Authors  underline  that difference in clinical presentation could be linked to a different pathophysiology but, as they add, unfortunately, all of the studies on ACS in their  meta-analysis did not specify the
pathophysiology , which is very important. There are many more ACS presentation in women, Microvascular angina, MINOCA; Takotsubo, coronary dissection in young women. I suggest, therefore, to give more importance to the limits and advantages of  method ( metanalysis) than to results which are inconclusive. 

Very important in my opinion it  is  to emphasize in the recent " gender semantic" what stressed by  DeVon et al ,mentioned in references)  to stop 
using the terminology "typical" and "atypical"; this still  represents a stereotype in many articles on ACS and gender

Author Response

#Reviewer 2

Well done metanalysis and this is crucial. In relation to be an original contribution, nothingis  really new about ACS symptoms in women. Authors  underline  that difference in clinical presentation could be linked to a different pathophysiology but, as they add, unfortunately, all of the studies on ACS in their  meta-analysis did not specify the
pathophysiology , which is very important. There are many more ACS presentation in women, Microvascular angina, MINOCA; Takotsubo, coronary dissection in young women. I suggest, therefore, to give more importance to the limits and advantages of  method ( metanalysis) than to results which are inconclusive. 

Very important in my opinion it  is  to emphasize in the recent " gender semantic" what stressed by  DeVon et al ,mentioned in references)  to stop 
using the terminology "typical" and "atypical"; this still  represents a stereotype in many articles on ACS and gender

Response: Following the reviewer's comments, we resumed the discussion by re-emphasizing the strengths of the meta-analysis. We also emphasized the cessation of this typical or atypical terminology. We thank the reviewer for allowing us to improve this work.